# Physical Laws Shape Up HOX Gene Collinearity

**DOI:** 10.3390/jdb9020017

**Published:** 2021-05-06

**Authors:** Spyros Papageorgiou

**Affiliations:** Institute of Biosciences and Applications, National Center for Scientific Research ‘Demokritos’, 153 10 Athens, Greece; spapage@bio.demokritos.gr

**Keywords:** hox gene collinearity, spatial temporal collinearity, vertebrates, Noether theory

## Abstract

Hox gene collinearity (HGC) is a multi-scalar property of many animal phyla particularly important in embryogenesis. It relates entities and events occurring in Hox clusters inside the chromosome DNA and in embryonic tissues. These two entities differ in linear size by more than four orders of magnitude. HGC is observed as spatial collinearity (SC), where the Hox genes are located in the order (Hox1, Hox2, Hox3 …) along the 3′ to 5′ direction of DNA in the genome and a corresponding sequence of ontogenetic units (E1, E2, E3, …) located along the Anterior—Posterior axis of the embryo. Expression of Hox1 occurs in E1, Hox2 in E2, Hox3 in E3, etc. Besides SC, a temporal collinearity (TC) has been also observed in many vertebrates. According to TC, first Hox1 is expressed in E1; later, Hox2 is expressed in E2, followed by Hox3 in E3, etc. Lately, doubt has been raised about whether TC really exists. A biophysical model (BM) was formulated and tested during the last 20 years. According to BM, physical forces are created which pull the Hox genes one after the other, driving them to a transcription factory domain where they are transcribed. The existing experimental data support this BM description. Symmetry is a physical–mathematical property of matter that was explored in depth by Noether who formulated a ground-breaking theory (NT) that applies to all sizes of matter. NT may be applied to biology in order to explain the origin of HGC in animals developing not only along the A/P axis, but also to animals with circular symmetry.

## 1. Introduction

Hox Gene Collinearity (HGC) is a basic embryonic property, coordinating the development of vertebrates and many other animal phyla. It was first observed by E.B. Lewis in the *Drosophila BX-C* gene complex [1]. Lewis noticed that a class of Hox genes is located in clusters following an ordered sequence (Hox1, Hox2, Hox3, etc.) along the direction 3′ to 5′ on the chromosome (Figure 1). These genes are expressed in the same order in the embryo along the anterior/posterior (A/P) axis. This common order in the chromosome and the embryo is denoted spatial collinearity (SC). SC is a surprising property because it correlates spatial entities differing by about four orders of magnitude. The linear dimension of an early vertebrate embryo is about 1mm, whereas the linear dimension of a Hox cluster is about 100 nm [2]. Biomolecular mechanisms, by themselves alone, cannot explain such interactions. Some long-range force must be involved. This is a general feature in Nature when phenomena at different ranges interact. The nature of the forces may differ, e.g., diffusion, electric forces, etc. For instance, in the hydrogen atom, a Coulomb force holds the electron at a long distance away from the nucleus (proton) in a fixed Bohr orbit.

SC is not the only property controlling Hox gene expression in vertebrate embryonic growth. Another principle for vertebrate development was later established: temporal collinearity (TC) [3,4]. According to TC, the time dependence of Hox gene expressions follows the empirical rule: in the ordered sequence of Hox genes from the telomeric to the centromeric end of the Hox cluster, there is a corresponding sequence of embryonic ontogenetic units E1, E2, etc. along the anterior–posterior axis. First, Hox1 is expressed in E1, followed later by Hox2 expressed in E2, etc. [3,4]. This is a schematic picture (for a more realistic description of Hox gene expressions see Section 6).

In the evolutionary process, a Hox cluster may appear in several homologues as a result of whole genome duplications (WGD) [5]. For instance, in vertebrates, there are four homologue Hox clusters, denoted HoxA, HoxB, HoxC, HoxD. These homologue clusters cooperate for normal embryonic development. The ordered Hox genes of a cluster (Hox1, Hox2, etc.) constitute a paralogy group (PG) [4,5]. For instance, Pg1 can be traced in Hox clusters of different animals as a result of their origin from a common ancestor Hox1. In vertebrates, each cluster contains a number of Hox genes belonging to 13 paralogues (Pgs) whose role seems to be crucial, since they have been traced in several vertebrates (see Section 4). In many cases, due to whole genome duplications, the genomes contain several copies of the Hox clusters.

In a comprehensive study of Hox gene expression in *Xenopus laevis*, M. Kondo et al. analyzed the cluster structures before and after allotetraploidization, which occurred 17–18 million years ago [5]. Recently, the whole genome sequencing of *X. laevis* was completed, and it is possible to compare the Hox clusters of the clawed frogs. The ancestral genome consists of two subgenomes, L (long) and S (short); most of the L and S subgroups are present in the genomes after allotetraploidization [5]. The two copies are designated as ‘homologs’ with suffixes L and S. For the four homologous gene clusters (HoxA, HoxB, HoxC, HoxD) there are eight ‘homolog’ clusters (HoxA.L, HoxA.S, HoxB.L, HoxB.S, HoxC.L, HoxC.S, HoxD.L, HoxD.S). Polyploidy (from 2 to 12 chromosome copies) is a characteristic feature of *X. laevis.* This feature makes *X. laevis* suitable to study gene duplication in vertebrates. In their comprehensive analysis, M. Kondo et al. noticed that many Hox gene expressions are consistent with TC (Hox genes with a higher Pg are expressed later than genes with a lower Pg) [5]. However, there are some genes violating TC. They suggested that the TC hypothesis must be revisited by comprehensive analysis of the developmental timing of transcriptional initiation of Hox genes in *X. laevis* and compared to *X. tropicalis* relevant data when they will be available [5].

From a different point of view, the doubt of TC validity was challenged by Durston et al., indicating that TC is verified in many chordates, including cephalochordates [6,7]. In particular, Durston observed that, in *Xenopus,* TC starts early at the non-organizer mesoderm and it is converted to a dorsal pattern (spatial), then he formulated a Time-Space-Translation Hypothesis [7]. In cooperation with this time-space-translation, TC leads to the development of different body parts of the vertebrate embryo [7].

In view of the above doubt of TC validity, it is worth interpreting the recent contradicting data by applying a Biophysical Model (BM), which provides a unifying approach.

## 2. The Biophysical Model for Hox Gene Activation

The basic hypothesis is that pulling forces act on the Hox clusters, and these pull-ing forces are influenced by contributions from both the microscopic and macroscopic scales. At the embryonic (macroscopic) scale, the contribution is contained in a morphogen gradient along the anterior-posterior embryonic axis [8,9,10]. The contribution from the microscopic scale originates from the cluster itself. Before activation, the Hox cluster is packed inside the chromatin territory (CT) in a compact unity (Figure 2). No forces are created at this stage. Gene activation starts when attractive forces are gradually created. Such forces emerge when polar molecules (positively charged) are transferred and allocated at the telomeric region of the cluster. (Detailed experimental evidence for these events is found in [11]). The forces gradually increase, pulling the genes out of the CT (Figure 2). Tentatively, the nature of the forces is quasi-Coulomb electric [8,9].

A proper Coulomb force Fc is defined:Fc = [(q1) × (q2)]/[R^2^]
where q1 and q2 are the electric charges of the electric bodies (positive or negative), and R is their relative distance. In our case, the proposed heuristic force F is a truncated Coulomb force, since the R^−2^ dependence is missing. It turns out that this guess works properly in explaining the data. F is the measure of the heuristic forces, and the arrows indicate their direction (Figure 2). A simple heuristic form for the attractive forces **F** acting on Hox clusters is the following [10,11]:F = N × P(1)

F consists of two factors: N, which represents the negative charge of the DNA of the Hox cluster, and P, the positive charge of the allocated molecules opposite the cluster (Figure 2). P are hypothetical molecules. It is legitimate to do this, since F does not contradict any First Principle. (About 50 years ago, the morphogens were also hypothetical). In the schematic Figure 2, the genetic (microscopic) range of N and the embryonic (macroscopic) range of P are indicated.

In Equation (1), factor N represents the contribution of the Hox cluster, which is negatively charged in agreement with the overall negative charge of DNA. The positive factor P is graded (low anteriorily, high posteriorily) and represents the embryonic contribution to F. The force F is an electric quasi-Coulomb force and applies at the telomeric end of the Hox cluster (Figure 2). In BM, the attractive force F provides an inter-play between the microscopic scale of the Hox cluster and the macroscopic embryonic scale (Figure 2).

## 3. The Irreversibly Expanding Spring Approximation

With the development of novel technological methods like super-resolution imaging of stochastic optical reconstruction microscopy (STORM), it is now possible to measure the geometric modifications of Hox clusters during Hox gene expression [12,13,14]. It was thus found that the Hox clusters are gradually elongated across the 3′ to 5′ axis of the cluster during gene activation. This observation supports the hypothesis that the Hox clusters behave like an expanding elastic spring. BM predicts this behavior, which is further supported by experimental evidence [15,16,17] (Figure 2). Following an increase of the morphogen gradient (from head to tail), the pulling force F increases, and the number of extruded Hox genes increases accordingly.

*(A)* 
*The Traditional Approach*


The mechanical properties of the expanding spring follow Hooke’s physical law which states that, for a wide range of pulling forces, the elongation of the spring is proportional to the measure of the force F, which is applied to one of the spring’s ends (Figure 3). At the other end of the spring, the spring is fastened. For the spring’s proper function, besides the pulling force F, the spring’s fastening is equally important. In the case of the mouse HoxD cluster, the gene regulatory region of the cluster plays the role of the spring’s fastening domain. For simplicity, the role of the ‘stiffness’ of the spring is ignored together with the local interactions of the constituent chromosomal configuration [18]. In the normal case of animals with *wildtype* development, the spring is completely fastened. If no force is applied on the spring, the spring remains uncharged.

The traditional tools and methods to explore the genetics of gene clusters are the chemical analyses of the biomolecules involved. This traditional methodology in Hox gene research has been combined with genetic engineering techniques like DNA excision or duplication and the subsequent biomolecular analysis of the expression modifications of the neighboring Hox genes (see e.g., [19,20]).

According to experiments performed with the above traditional techniques, it is now established that the regulatory elements of the mouse Hox clusters are posteriorιly located, upstream of the cluster even beyond gene Evx2 (Figure 1B). A detailed search has explored this upstream area. Partial excisions of this area cause significant modifications of Hox gene expressions compared to *wildtype* expressions [21].

If the force F is weak, the spring will be slightly shifted, while the spring fastening is complete (Figure 3B). If the fastening is partly relaxed, by removing part of the gene regulatory region, the same weak force F will slide the spring further (Figure 3C). If the fastening is completely removed, F will shift the spring even further (Figure 3D). The above picture leads to a BM prediction: the total removal of the gene regulatory region automatically causes the shifting of the whole cluster in the gene activation region and no gradual Hox gene expressions should be observed.

The above expectation is a BM prediction in retrospect: amazingly, as early as 1999, T. Kondo and D. Duboule observed this phenomenon, since HoxD4 and HoxD10 expressions appear earlier at a time corresponding to that of HoxD1 appearance ‘as if temporal collinearity disappeared’ ([21], p. 414). Therefore, the expanding spring approximation directly relates the complete deletion of the gene regulatory elements of the HoxD cluster with the disappearance of TC as a result of the early movement of the whole cluster inside the transcription factory domain.

*(B)* 
*A Novel Approach Involving Conserved Non-Coding Elements (CNE)*


In the last 20 years, a novel powerful weapon has been added to the tools exploring the properties and activation of Hox gene clusters. Besides the protein coding genes in the genome, there are more than 30 thousand RNA elements in the human genome which do not code any proteins (ncRNA). More than thousand of these non-coding RNAs are persistently conserved from generation to generation [22]. Lately, the conserved DNA non-coding elements (CNE) and their long non-coding RNAs (lncRNA) have been intensively studied by many groups with the help of sophisticated numerical analysis methods [23,24]. Their findings are impressive. The numerous CNEs are preserved for more than 400 million years of evolution. The size of these CNEs varies; it can reach more than 300 bp. The CNEs play different roles in normal development and disease, coordinating spatial-temporal gene expression in both embryos and adult animals. A CNE that has been thoroughly studied is *Hotair*, which is a lncRNA located between HoxC11 and HoxC12 in the vertebrate chromosome 12 [24]. Its location in the posterior domain of the HoxC cluster is compatible with the hypothesis that CNEs can be involved in the creation of the pulling forces of BM. In what follows, it is indicated that CNEs play an important role, particularly in the expanding spring approximation. It is further assumed that several CNEs are located in the fastening posterior domain of the vertebrate Hox clusters.

Some early studies of CNEs in humans and mice were reported in 2011 [25]. Recent findings show that deletion of the *Hotair* locus has no detectable effect on HoxD genes in vivo [26].Furthermore, the mouse Hotair knock-out causes derepression of HoxD genes [27]. Even if *Hotair* shows low sequence conservation in several vertebrates, it has been noticed that many ncRNAs are conserved in structure, although not conserved in sequence [28].

In view of the above findings, an International Bioinformatics Group in collaboration with the Imperial College team headed by B. Lenhard have recently performed a comprehensive analysis of the regulatory roles of *Hotair in cis* and *in trans* on Hox clusters [24]. These authors propose that at the second round of whole genome duplication, human HOTAIR expression is correlated positively with HoxC11 *in cis* and negatively correlated with HoxD11 *in trans* [24]. They compared human and zebrafish CNEs and identified a 32-nucleotide long CNE conserved across the vertebrates. They characterized this long CNE as the ancestral sequence of the ancestral HoxC/D cluster. Thus, they confirmed the Nepal et al. proposition [24]. This is a challenging hypothesis to be further tested. More specifically, this analysis indicates that HOTAIR expression regulates human HoxC11 *in cis* positively and HoxD11 *in trans* negatively [24]. The number of the conserved elements varies, depending on the cluster copies during the whole genome duplications. Therefore, a certain pulling force will cause a variable shift (sliding) of the cluster toward its telomeric end, and the starting time of Hox gene expression will be accordingly modified.

In the expanding spring approximation (Approach A), the regulatory gene domain controls the cluster fastening [15,17]. In the novel Approach B, the fastening is achieved with the action of a set of CNEs. Note that both approaches lead to comparable results.

As stated in the Introduction, in the case of *X. laevis*, M. Kondo et al. analyzed the Hox gene expressions and concluded that the initiation of these expressions deviates from the normal paralogue Pg order of Hox gene expressions; therefore, they violate TC [5]. Following the WGD, the number of CNEs fastening the Hox clusters may vary so that the total effect on Hox cluster expansion differs for the various clusters. According to Approach B, the initiation and duration of gene activation is reshuffled. This could explain the violation of TC in the L and S homologs of the *X.leavis* as observed by M. Kondo et al. [5].

## 4. Symmetries in a (Finite) Linear Ordering

Symmetry is a very broad concept with many facets covering physical, mathematical, aesthetic and philosophical aspects. Many appropriate definitions have been proposed [29,30,31]. For the present purpose, the compact definition by Wilczek in the form of aphorism is preferred: symmetry is ‘change without change’ [31]. For example, consider a circle in a plane and a perpendicular axis passing through its center. Any rotation around this axis is an operation that moves any point of the circle to some other point of the circle so that the circle remains invariant. This very simple and obvious example is the start of a far going line of thoughts that Emmy Noether followed in 1918 and discovered a fundamental principle of Mathematics and Physics with philosophical repercussions. Noether started with Classical Mechanics and proved rigorously that a physical system obeying a symmetry law is necessarily followed by a conserved quantity. As an example, Noether proved that a physical law symmetrical under spatial rotations (freedom to choose any orientation for the coordinate system) leads to the conservation of angular momentum [30,31]. A simple and exact exposition of this Noether Theory (NT) is found in [32]. (For the significance of Symmetries and NT, see the Appendix A).

Of particular interest here is the symmetry of multi-scale objects or phenomena. A class of such objects are the fractals [33]. The symmetry of fractals is called self-similarity and it characterizes the property of objects being similar to each other but in different scale. In Nature and Life, self-similar entities are quite frequent. For example, self-similar is the pattern of branchings of the blood vessels in the lung or the multi-scalar Barnsleyleaf [33] (Figure 4). In the original (theoretical) formulation of self-similarity, the scale variation is continuous (and infinite), whereas in HGC, only two spatial scales are involved, the embryonic and the Hox gene cluster scales. Nevertheless, in this primitive case of symmetry, it is tempting, according to NT, to expect a primitive conservation of some corresponding quantity. In the common ancestor Hox gene cluster, by inspection, the complete ordering of Hox genes (Hox1, Hox2, Hox3, …, Hox13) is conserved (Figure 1A). For vertebrates, in the homologue Hox clusters (HoxA, HoxB, HoxC, HoxD) the ordering is not complete: some genes are missing, while the Pg order is preserved. It is believed that, together with the WGD and the evolutionary process, some genes faded out (gene loss), and all four homologue clusters cooperated with specific roles.

Amphioxus is a close relative of vertebrates and a direct descendant of the vertebrate ancestral existing before WGD. It possesses a 14th Hox gene without any gene loss; therefore, it is suitable to study the evolutionary history of vertebrates [34]. Furthermore, a numerical comparison study of *C. amphioxus* and mouse Hox clusters showed that BM is consistent with vertebrate Hox clusters, adopting more involved organization as a tool to develop more complex body structures during evolution [35].

By inspection of the existing *wildtype* vertebrate Hox gene data, a partly conserved piecewise Hox gene ordering is observed, which follows the rule: Pg is irreversibly increasing like a ‘ratchet’, even if some genes are missing. For instance, the *wildtype* Pg ordering (1, 2, . , . , 5, 6) is allowed. In contrast, the ordering (1, 2, . , . , 6, 5) is forbidden because it represents a gene reversal mutation which is not allowed [36].The abnormal spontaneous mutant of *Antennapedia* in the *Drosophila* is an example where, in the location of antennas, legs are growing as a result of an abnormal reversal of the corresponding Hox genes [37]. These mutations are named *Homeotic.*

## 5. Symmetries in a Circular Gene Ordering

The Symmetries of Section 4 underlie the description of organisms whose embryos grow along the A/P axis, retaining the same pattern in their adult life, e.g., arthropods or vertebrates. The embryos of these ‘directly’ developing animals look like miniatures of the adult organisms. Besides these animals, there is a large variety of invertebrates which, at the very early stages of embryogenesis, are ‘indirectly’ developing a larva next to the body of the embryo where the pattern of the adult animal is formed. This body pattern differs substantially from the initial embryo organization along the linear A/P axis [38,39,40,41]. *Strongylocentrotus purpuratus* is a typical indirectly developing sea urchin whose larva organization is circular (Figure 5).

The first genome sequencing data of the echinoderms were published in 2006, and it was followed by many others [39,40]. The sequencing was unexpected, and several models have been proposed to explain the data [40,41]. However, many questions still remain unanswered [42].

In what follows, another ordering mechanism is proposed based on the symmetries of the preceding Section. According to this hypothesis, the initial linear Hox gene organization is transformed into a circular Hox gene reorganization at the larva stage [42]. (Figure 5). (This organization change has been observed in some vertebrates). Double strand break (DSB) is a mechanism used in experiments of DNA rearrangements to cure serious illnesses like cancer, but it is also a tool in spontaneous DNA reshufflings, leading to evolutionary novelties. In a recent review, both experimental and spontaneous DSB were extensively treated [43]. In hundreds of millions of years of evolutionary history, DSB has played a critical role in the pathway from simple ontological structures to more complex entities [35].

In Figure 6A, a diagram is depicted where, at the larva stage, the Hox genes of a sea urchin cluster are bended, forming a loop, conforming to the circular organization of the ontological units of the embryo (Figure 5A). Schematically, the last Hox13 at the posterior end 5′ of the cluster is approaching Hox1 at the anterior end 3′ in the area where the DSB occurs (in the elliptic disc).

In every Hox gene of the circle, a ‘circular’ identity is imprinted which differs from the initial linear identity. The term ‘circular identity’ is vague since almost nothing is known about it. It is however useful, since it fits well in a solid theoretical (hypothetical) background. If Hox1 is connected to the 3′ end of the flanking chromosome (and Hox13 to the 5′ end), no novel DNA sequence is created. Although a circularly symmetric invertebrate, *A. planci* retains the Hox ordering of directly developing animals. This is due to the insertion pathway of the cluster in the flanking DNA: Hox1 is connected to the 3′ end and Hox13 to the 5′ end of DNA. In such insertion, no novelty is created (See Figure 6 and Figure 7B).

In contrast, a novelty will arise if Hox1 is connected to the 5′ end of the chromosome (and Hox13 to the 3′ end) (Figure 6C). This is the case of the *sea urchin*, where the gene order is the result of an abnormal Hox cluster insertion in the flanking DNA (Hox1 to the 5′ end and Hox13 to the 3′ end), and a novelty is created (Figure 6C). Furthermore, a **second** DSB can occur at the missing Hox4 location (between Hox3 and Hox5). The two ends of the remnant (smaller) Hox cluster (Hox5 and Hox13) are correspondingly attached to the two ends 3′ and 5′ of the flanking DNA (Figure 6C). This leads to the final *sea urchin* gene ordering shown in Figure 7C.

At this point, it is more realistic to represent the linear Hox cluster ordering of Figure 1 and Figure 6 as a two-dimensional strip. This dimensional change causes a deep topological modification. If this strip is bent so that the two ridges come close to each other, a cylindrical surface will be formed (Figure 8A) [45]. The bending around is performed for 360° and the Hox genes obtain a new circular configuration. If the cylindrical surface is opened, the system returns to its original two-dimensional strip, while the genes retain their circular identity. At the same time, another operation can be performed: if one of the strip’s edge is twisted 180° and then joined to the other edge of the strip, a two-dimensional loop will be created, embedded in the 3-dimensional space. The created loop is an endless one-sided surface called ‘Moebius strip’ (Figure 8B) [45]. The 2-dimensional twisting of the strip is an extension of the one-dimensional (abnormal) connection of Hox1 to the 5′ end of the flanking DNA (and Hox13 to the 3′ end), as shown above in Figure 6.

This procedure (of bending and twisting) may be extended to three-dimensional surfaces. A multiple turning around 360° with one twisting creates a multiple Moebius strip in the shape of a ring, the so-called ‘Moebius torus’ (Figure 8C). If additional twists are performed, more complex structures are created, suitable to host several invertebrate genomes with variable symmetries (Figure 6) [45]. As an example, a Moebius torus can accommodate a sequence of Hox clusters in the chromosome or the genome of a starfish with five podia and five twisting (Figure 5B).

## 6. Conclusions

The second half of 20th century was an exciting time for developmental biology. Although the discovery of DNA structure was a historical turning point for all branches of Biology, the understanding of many local molecular mechanisms was not equally satisfactory. For instance in 1969, the pivotal work of L.Wolpert and the ‘French Flag Problem’ were quite vague and abstract with no molecular verification [46,47]. Nevertheless, the notion of morphogens and their gradients were fruitful in guiding research to the right direction. Since then, spectacular technological advancements, hand in hand with basic research, have promoted our knowledge to an unbelievable depth. For instance, the contemporary status of development and, more specifically, the level of our knowledge of Hox genes and the role they play in both biology and medicine.

It has been suggested that TC is the ‘principal constraining force’ keeping Hox clusters in a compact organization [48]. In the case of a complete deletion of the regulatory region, BM predicts that the Hox cluster becomes completely loose; therefore, it can move like a freely moving body [49]. A small pulling force (e.g., at a very early stage of activation) initiates the automatic cluster sliding toward the transcription factory domain. Consequently, no gradual activation of the Hox cluster is possible, and TC disappears [21] (see Section 3). The above BM prediction leads to a daring hypothesis for the absence of temporal collinearity in *Drosophila*. The mechanism of TC disappearance in *Drosophila* could be the same mechanism predicted above by BM in combination with the evolutionary destruction pathway from an ancestral Hox cluster of ‘*organized type (O)’* [4] to a Hox cluster lacking TC [21].

In Figure 1, the distances between Hox genes and their expressions are depicted as small squares and circles. This is only schematic and only the 3′ end of the gene expressions are clearly observed. The 5′ end of these expressions is smeared out so that, in many cases, there is an overlap of the cluster gene expressions. It has been further noticed that in cells where such an overlap occurs, the expression intensity of a Hox gene follows its Pg order. For instance, in the same cell, the expression intensity of a Hox11 gene is stronger compared to the intensity of Hox10.

This phenomenon is called ‘quantitative collinearity’ [9]. (For the overall significant role of Pgs, see the Appendix B).

An application of BM could be useful in cancer research. As stressed above, in the normally developing vertebrate embryos, the pulling forces gradually elongate the Hoxcluster [17,49]. It has been observed in many cases that overexpression of Hox clusters are concurrent with myelodysplastic syndrome [50]. In other cases, overexpression of HoxA and HoxD were found in ovarian cancer with unknown etiology [51]. Here, a start of a possible explanation is proposed: partial or total deletions of the regulatory region of Hox clusters can cause abnormal cluster elongations (Section 3). These abnormal elongations are related to Hox gene over-expressions. With the plethora of data accumulated in the existing Big DataBases, it is probably a matter of ‘clever digging’ in these data to confirm (or reject) a correlation between mutations in the regulatory region of Hox clusters on one hand and on the other, the abnormal Hox cluster elongations combined with Hox gene overexpression and specific forms of malignancy [51].

## Figures and Tables

**Figure 1 jdb-09-00017-f001:**
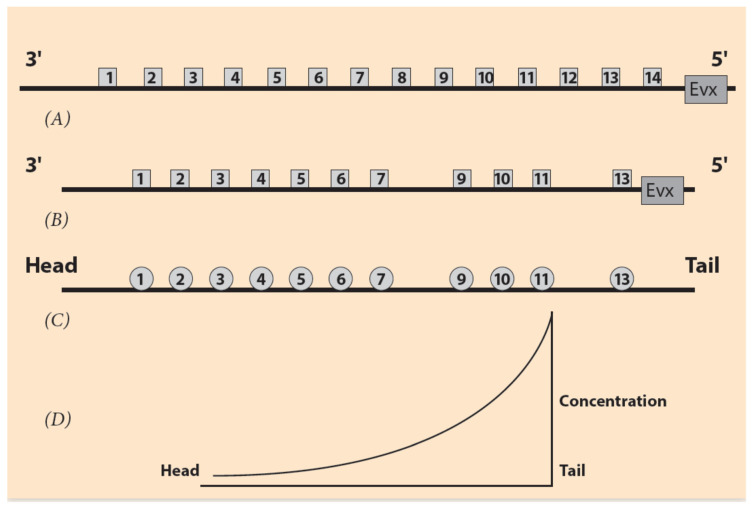
Ordering of Hox genes and the sequence of the ontogenetic units. (**A**) Hox gene ordering of a (theoretical) common ancestor. Gene **Evx** is located next to the 5′ end of the Hox cluster. (**B**) Ordering of the mouse HoxA cluster. Hox8 and Hox12 are missing. (**C**) The corresponding ontogenetic units of the mouse along the A/P axis. (**D**) The steady state monotonic concentration gradient of a morphogen. The peak is at the tail region.

**Figure 2 jdb-09-00017-f002:**
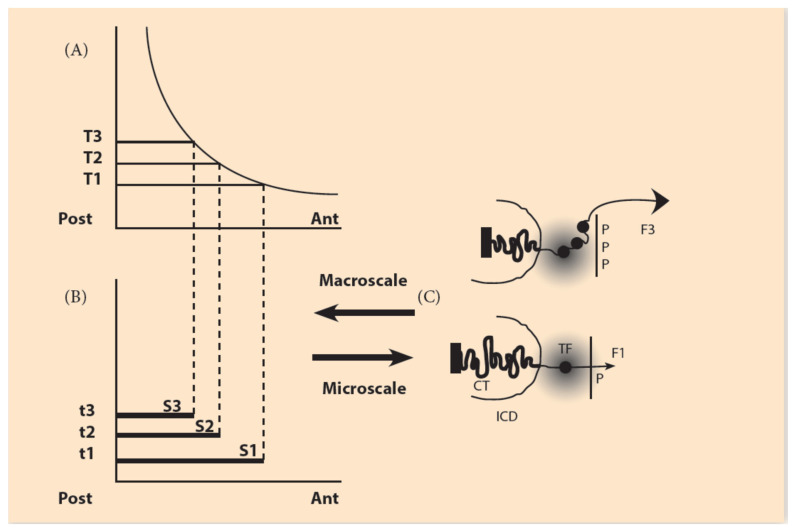
The macroscale morphogen gradient and the microscale Hox gene clustering in space and time (adapted from Papageorgiou S. (Biology 2017: 6, 32)). (**A**) Concentration thresholds (T1,T2,T3) divide the A/P axis in partially overlapping expression domains. (**B**) The time sequence (t1, t2, t3) combined with the thresholds sequence (T1, T2, T3) determines the Hox1, Hox2, Hox3 activation in space and time. S1, S2, S3 are the partially overlapping and nested expression domains of Hox1, Hox2, Hox3. (**C**) (bottom) In an anterior cell of S1, a small forceF1, pulls Hox1 (black spot) out of the chromatin territory (CT) toward the Interchromosome Domain (ICD) in the regime of the Transcription Factory (TF) (grey domain). Allocation of polar molecule P opposite the telomeric end of the Hox cluster (bottom). At a later stage (top), in a more posterior location of S3, a stronger force F3, pulls Hox1, Hox2, Hox3 out of CT in the TF (Allocation of 3P molecules).

**Figure 3 jdb-09-00017-f003:**
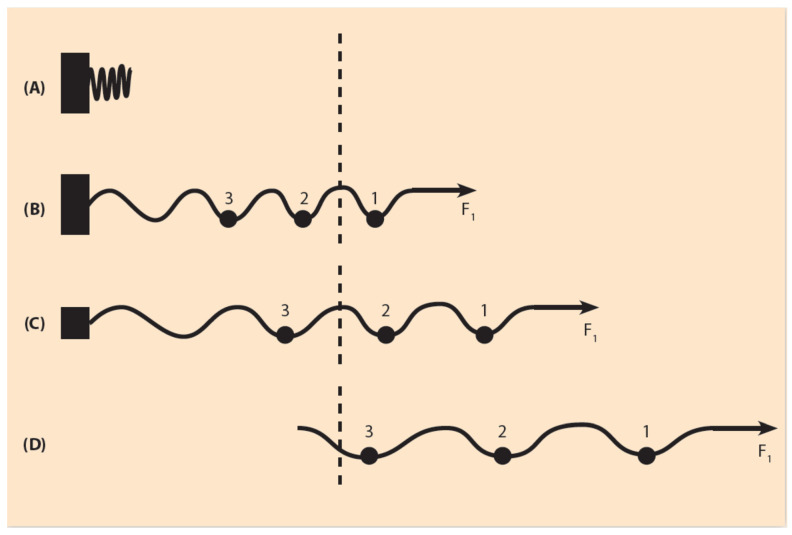
Elastic spring expansion under a small pulling force (schematic). (**A**) In the uncharged spring, no force is applied. The spring is compacted next to the fastening region. (**B**) A small force F1 is applied to the right end of the spring. The spring fastening is complete and the spring expands slightly. A small sphere 1 moves to the right beyond the dashed line where gene activation occurs. (**C**) The spring fastening is reduced (smaller black rectangle) and the spring, under F1, slides further to the right (two small spheres pass to the activation region). (**D**) The fastening is completely removed and, under the same force F1, all three small spheres 1, 2, 3 move into the activation region.

**Figure 4 jdb-09-00017-f004:**
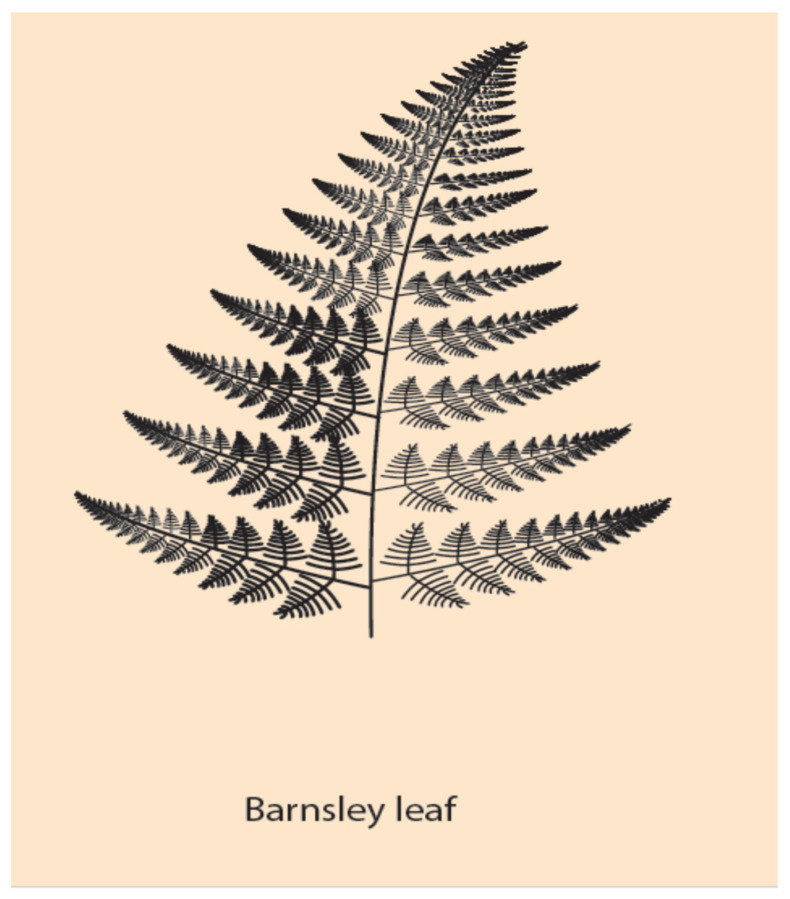
A self-similar design of a Barnsley leaf: Each branch of the leaf is similar to all other branches, differing only in their scale (bigger or smaller).

**Figure 5 jdb-09-00017-f005:**
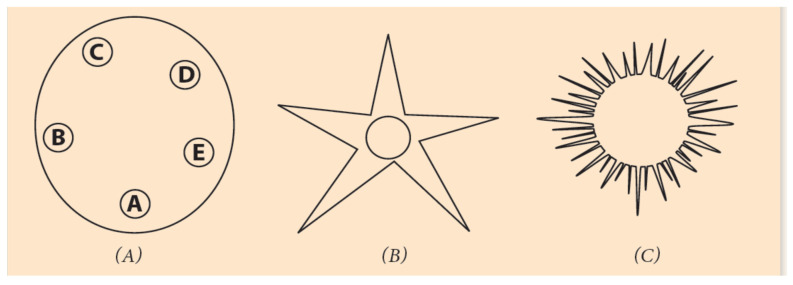
Circular organization of echinoderms at different developmental stages. (**A**) *Strongylocentratus purpuratus larva* with five podia. (**B**) An adult starfish with fivepodia. (**C**) A juvenile sea urchin.

**Figure 6 jdb-09-00017-f006:**
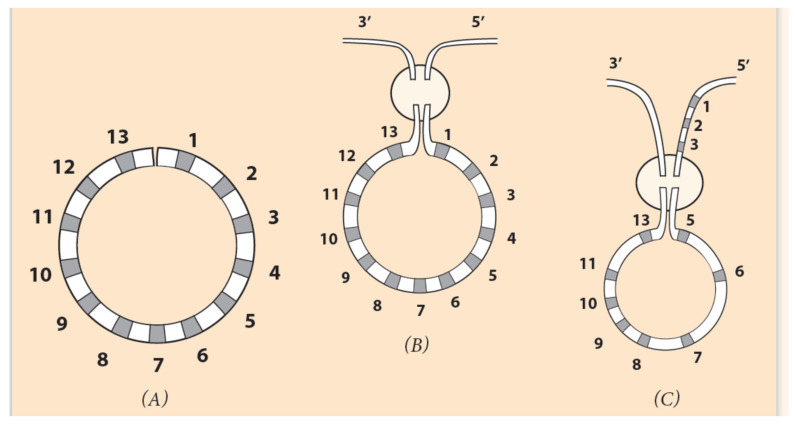
Circular ordering of Hox genes. (Adapted from S. Papageorgiou (2016) Current Genomics). (**A**) The linear DNA is bent, and the two ends of the Hox cluster come close together. (**B**) In the encircled domain, the ends Hox1 and Hox13 of the cluster are connected to the 3′ and 5′ end of the flanking chromosome. If Hox1 is attached to the 3′ end and Hox13 to the 5′ end, the produced linear arrangement is the normal one and could represent the observed *A. planci* gene ordering. (**C**) If Hox5 is connected to the 3′ end and Hox13 to Hox3 on the flanking chromosome, the linear ordering is the *seaurchin* Hox cluster arrangement. (See explanation below.).

**Figure 7 jdb-09-00017-f007:**
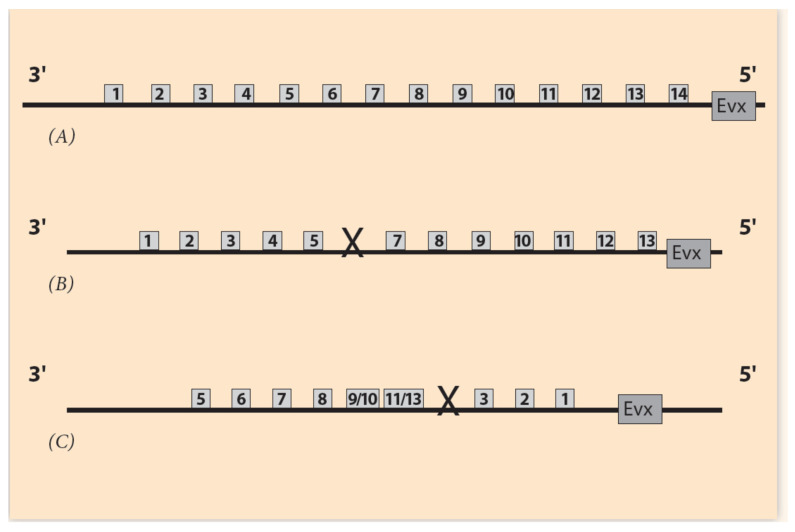
Comparison of Hox gene ordering of *Amphioxus*, *A. planci and sea urchin.* (**A**) Amphioxus gene cluster has a 14th Hox gene without any gene loss. (**B**) *A. planci* gene cluster with one gene loss at Pg6 where a DSB occurs [44]. (**C**) *Sea urchin* gene cluster with one gene loss at Pg4 position where DSB occurs.

**Figure 8 jdb-09-00017-f008:**
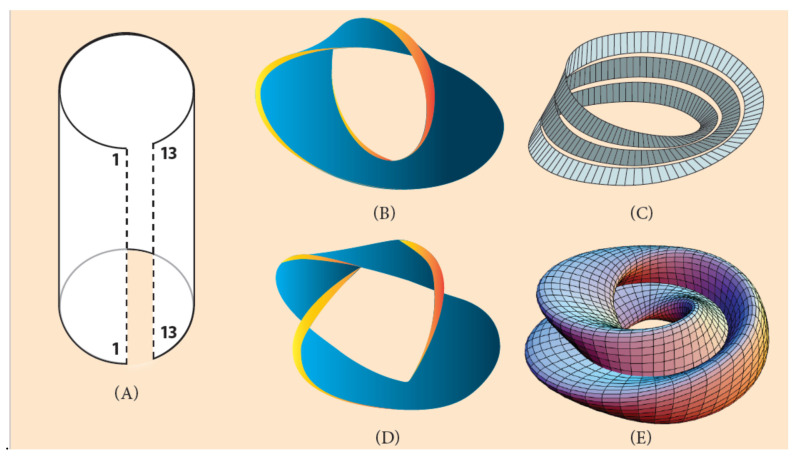
Moebius strip constructions (Adapted from Wikipedia). (**A**) A two-dimensional strip is turned around with the two edges 1/1 and 13/13 coming close to each other. If edge 1/1 is connected to the 3′ end of the flanking DNA and edge 13/13 to the 5′ end of DNA (look at the one-dimensional connection of (Figure 5B)), the gene order returns to the normal ordering of Figure 6B (*A. planci*). (**B**) A two-dimensional strip is turned around 360° and one 180° twisting (a Moebius circle). (**C**) A two-dimensional strip turned around 360^0^ three times and one twisting. (**D**) A two-dimensional strip twisted several times. (**E**) A Moebius torus.

## Data Availability

Not applicable.

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
