# Peer review of "Physical Laws Shape Up HOX Gene Collinearity"

_jdb, 2021, doi:10.3390/jdb9020017_

Round 1
Reviewer 1 Report
This is an interesting and comprehensive review on the recent development in elucidating the molecular mechanism underlying spatial and temporal collinearity of Hox genes during embryonic development. The author did a fine job introducing the concept as well as describing the recent exciting biophysical model explaining Hox gene activation. Overall, the manuscript is easy to read and follow. However, there are several places where better explanation may make the paper more reader-friendly.
Specific points:
- Can the author elaborate and/or speculate more on the nature of the pulling force F, does it have something to do with the molecular motor and/or nuclear matrix? Likewise the nature of P?
- Line 11, 12, 13, 15 and throughout the body of the text, it is better to use “et. Cetera” than ellipses “…”
- Line 13, “embryo” is misspelled as “embyo”
- Lines 14 and 15 should read “According to TC, first Hox1 is expressed in E1, later Hox2 is expressed in E2, followed by Hox3 in E3 et. cetera.”
- Use a sentence in the abstract to more clearly link the biophysical model with Noether theory
- Line32-36, the basis for this argument is not immediately clear to this reviewer. What is the point of comparing the linear dimension of a Hox cluster and the embryo? What does biomolecular mechanism refer to? Lines 35 and 36, citation is needed, or better explanation why biomolecular mechanisms cannot explain spatial collinearity across orders of magnitude. This section needs to be rewritten to better articulate the argument.
- Line 51 should read “These homologue clusters cooperate for normal embryonic development.”
- Line 53 should read “different animals”
- Line 54 should read “In vertebrates there are 13 paralogue groups (Pg1, Pg2 et. cetera through Pg13)
- Line 54, ...whose role is crucial. (for what?)
- The sentence in line 55 is already made clear at the beginning of the paragraph and is redundant
- Line 57 should read “analyzed WGD, and in particular an allotetraploidization event that they estimated to have occurred 17-18 million years ago [5].”
- Line 59 should read “L and S homologs are not ordered consistently with what would be expected according to TC”
- Line 71 should read “basic hypothesis is that pulling forces act on the Hox clusters, and the pulling forces are influenced by contributions from both the microscopic and macroscopic scales.
- Line 88, Apposition.....Hox cluster, as well as line 90- fragmentation, not a sentence.
- Line 130 should read “modifications of hox gene expression compared to wild type expression [21].”
- Line 145 in English we use comma to separate large numbers so please replace “30.000” with “30,000” or type out the whole word “thirty thousand”
- Lines 152, 237, 239, please replace “grown up” with “adult”
- Line 160 please replace “occurring” with “occur”
- Line 160-163, this section is very confusing and needs to be rewritten. what is the difference between line 162 and 163? The author needs to explain why they think the two studies have contradictory findings
- Line 161 “does not affect Hoxd transcription”
- Lines 169 and 174 are repetitive to one another.
- Line 201 please replace “Appentix” and “Appendix”
- Line 296 please replace “sustem” with “system”
- Line 356, Big Databases
Author Response
Dear Reviewer 1,
please find my comments below and attached the amended manuscript.
Manuscript ID jdb-1164186
Comments on Reviewer 1 Report
1 See new corrected version v2 (lines 80-82).
The nature of the BM pulling forces are quasi- electric. A proper Coulomb force Fc is defined :
Fc = [(q1)*(q2)]/[R2]
where q1, q2 are the electric charges of the electric bodies (positive or negative) and R their relative distance. In our case, the proposed heuristic force F is a truncated Coulomb force since the R-2 dependence is missing. It turns out that this guess works properly in explaining the data. F is the measure of the heuristic forces and the arrows indicate their direction. I shortly clarify this in the new version. F consists of two factors – N which represents the negative charge of the DNA of the Hox cluster and P the positive charge of the apposited molecules opposite the cluster (Fig. 2). P are hypothetical molecules.It is legitimate to do since they do not contradict any First Principle.Do not forget that, about 50 years ago, the morphogens were also hypothetical. In the schematic Fig. 2 the genetic (microscopic) range of N and the embryonic (macroscopic) range of P are indicated. In the new version (v3) a short explanatory description with references is added.
2 lines 11-13, 15 and thereafter I made the changes to et. Cetera.
3 line 13 Misprint corrected.
4 lines 14, 15 change to “According to TC, … et. Cetera”.
5 line 21 Sentence ‘ NT may be applied to Biology in order to explain’ added in the Abstract.
- lines 32-36 Section rewritten in the new version. L 32-36
explain such interactions. Some long-range force must be involved. This is a general rule in Nature when phenomena in different ranges are interacting. The nature of the forces may differ e.g. Diffusion, Electric Forces, et. Cetera. For instance in the Hydrogen atom a long-range Coulomb force holds the electron at a long distance away from the nucleus (proton) jn a fixed Bohr orbit.
7 line 51 Sentence corrected.
8, 9 lines53, Sentence accordingly corrected. ‘In vertebrates there are 13 paralogue groups Pg1, Pg2, et Cetera through Pg13).
10 line 55 ‘with every Pg playing a specific role in embryonic development
11 line 55 Sentence dropped.
12-13 Lines 57-63 are rewritten
14 line 71 Accordingly rephrased [basic hypothesis… macroscopic scales.]
15 line 88 Legend 2 corrected.
16 line 130 Sentence corrected
17 line 145 replacement of numbers
18 lines 152, 237, 239 replacement to adult
19 line 160 change to occur
20 lines 160-163 paragraph rewritten. Ref. [25, 26] concern to the work of the same group (Duboule’s Lab). Following Duboule et al. I adopt the follow-up version [26] and ignore the former [25].
21 line 161 omission of the
22 lines 168, 173 the second repletion is avoided.
23-25 lines 203, 287, 357 misprints corrected.
With best wishes,
Spyros
Reviewer 2 Report
All my comments are included within the paper document.

Author Response
Dear Reviewer ,
please find my comments below and attached the amended manuscript.
Manuscript ID: jdb-1164186
Comments on Reviewer’s 2 report
1 The nature of the BM pulling forces are quasi- electric. A proper Coulomb force Fc is defined :
Fc = [(q1)*(q2)]/[R2]
where q1, q2 are the electric charges of the electric bodies (positive or negative) and R their relative distance. In our case, the proposed heuristic force F is a truncated Coulomb force since the R-2 dependence is missing. It turns out that this guess works properly in explaining the data. F is the measure of the heuristic forces and the arrows indicate their direction. I shortly clarify this in the new version. F consists of two factors – N which represents the negative charge of the DNA of the Hox cluster and P the positive charge of the apposited molecules opposite the cluster (Fig. 2). P are hypothetical molecules.It is legitimate to do since they do not contradict any First Principle.Do not forget that, about 50 years ago, the morphogens were also hypothetical. In the schematic Fig. 2 the genetic (microscopic) range of N and the embryonic (macroscopic) range of P are indicated. In the new version a short explanatory description with references is added.
- line 97 N and P are crucial factors for the quasi-Coulomb force F. (See comment 1 above)
3 line 102 Evidence supporting the existence of F is given in ref. [12-14] where Duboule and co-workers apply novel high-resolution microscopy techniques. They observe gradual elongations of Hoxd cluster during Hox gene activation and they state explicitly that these observations are consistent and predicted by BM. Furthermore, Duboule et al. ‘predate’ the geometric restructuring of Hox clusters from transcription molecular processes[13] (2015) and BM predicts the ‘preceding’ of this happening one year before (http://webmedcentralplus.com/article_view/405) (2014).
4 The term ‘gene regulatory region’ is used here following ref.[19] as the regulatory element located upstream of the Hox cluster (including Evx2) necessary for setting up the early pattern of Hox gene collinear activation.
5 I think the Moebius strip approach besides being more realistic it helps describing more complicated gene (and ontological) configurations incorporating several chromosome bendings and twistings (Fig. 8). For instance, I am thinking if one can construct a chromosome corresponding to a starfish (Fig. 5B) ??.
6 I understand you do not agree to include the Appendices in the manuscript. I prefer to incorporate them in a spirit of scientific First Principles covering all topical issues like HGC.
With best wishes,
Spyros

Round 2
Reviewer 2 Report
The revised version is much clearer. My points have been fully addressed by the author. Just make sure that the final proof has adequate format (many words, in the revised version (pdf), seem to lack separation and others, like “et.Cetera” are not correctly spelled). Please, check the whole document for typographic errors. Otherwise I find the study interesting and a nice contribution to modelling the function of the HOX cluster. Needless to say, we should wait for more experimental data in order to fully test the model’s assumptions. Congratulations.
Author Response
The format and the typos are revised.